# Risk-Taking Behavior among Suicide Attempters

**DOI:** 10.3390/jcm11144177

**Published:** 2022-07-19

**Authors:** Nasrin Abdoli, Nader Salari, Vahid Farnia, Mehdi Khodamoradi, Somayeh Jahangiri, Masoud Mohammadi, Annette Beatrix Brühl, Dena Sadeghi-Bahmani, Serge Brand

**Affiliations:** 1Substance Abuse Prevention Research Center, Health Institute, Kermanshah University of Medical Sciences, Kermanshah 6719851115, Iran; abdolinasrin511@yahoo.com (N.A.); n_s_514@yahoo.com (N.S.); vahidfarnia@yahoo.com (V.F.); mehdi0khodamoradi@gmail.com (M.K.); jahangirisomayeh84@gmail.com (S.J.); 2Department of Biostatistics, School of Health, Kermanshah University of Medical Sciences, Kermanshah 6719851115, Iran; 3Sleep Disorders Research Center, Kermanshah University of Medical Sciences, Kermanshah 6719851115, Iran; 4Cellular and Molecular Research Center, Gerash University of Medical Sciences, Gerash 7441758666, Iran; masoud.mohammadi1989@yahoo.com; 5Center for Affective, Stress and Sleep Disorders, Psychiatric Clinics of the University of Basel, University of Basel, 4002 Basel, Switzerland; annette.bruehl@upk.ch; 6Department of Psychology, Stanford University, Stanford, CA 94305, USA; bahmanid@stanford.edu; 7Division of Sport Science and Psychosocial Health, Department of Sport, Exercise and Health, University of Basel, 4052 Basel, Switzerland; 8School of Medicine, Tehran University of Medical Sciences (TUMS), Tehran 1417466191, Iran

**Keywords:** risk-taking behavior, suicidal behavior, suicide attempts

## Abstract

Background: Suicidal behavior is a major mental health concern both for the individual and for the public health. Among others, suicidal behavior is associated with impulsivity, risk taking, pain tolerance, and a state of overarousal. In the present study, we investigated if suicide attempters (SAs) reported higher scores for risk-taking when compared with healthy controls (HC) of the general population. Methods: A total of 616 individuals (mean age: 27.07 years; 51.5% females) took part in the study; of those, 240 (39%) were individuals with a suicide attempt (SA) within a time lapse of one to three months, and 376 (61%) were healthy controls (HC). Participants completed a series of self-rating questionnaires covering sociodemographic information, risk-taking (Risk-Taking Questionnaire 18; RT-18), and suicidal behavior (Suicide Behaviors Questionnaire-Revised; SBQ-R). Results: Compared with HCs, individuals with SA reported higher risk-taking and suicidal behavior scores. The risk-taking questionnaire yielded a four-factor solution: Thrill and sensation seeking; Cautious procedure; Cautious decision making; Impulsive behavior. Compared with HCs, SAs showed the highest scores for thrill and sensation seeking and impulsive behavior. Conclusions: Compared with healthy controls, individuals reporting a recent suicide attempt also reported a higher propensity to thrill and sensation seeking and impulsive behavior as a proxy of risk-taking behavior. The present results corroborate the notion that, among others, suicide attempts appeared to be less related to premeditation, but rather to impulsive and thus spontaneous behavior.

## 1. Introduction

Suicide and suicidal behavior do not fit with the everyday experience to care for life and for our social bonding. While suicide is the decision deliberately to end one’s life, the suicide attempt is considered a failed suicide. Suicide is one of the leading causes of death in the world [1], and the tenth most common cause of death in the US [2]. Suicide and suicide attempts may lead to disturbing and long-lasting negative consequences both for the individuals concerned (in case of attempted suicide) and for the social environment, including family members, workplace colleagues, and witnesses to the suicide [1]. Further, subjectively to the individual with suicidal ideations, death appears to be the only solution to cope with a perceived unbearable life situation [3]. As such, suicide and suicidal behavior might be considered as a lack of alternative coping strategies.

To explain suicide and suicidal behavior, several concepts have been advanced (see [4,5,6,7] for extensive overviews). For clarity, such explanations could be synthesized to economical, physiological, social, and cognitive–behavioral clusters; although, plausibly, such factors contribute jointly to suicidal behavior. 

First, economic factors such as extreme poverty [8,9] and economic crises [10,11,12] might increase the risk of suicidal behavior.

Second, psychiatric disorders [13] such as bipolar and unipolar mood disorders [14,15,16,17,18,19,20,21,22,23,24], sleep disorders [25], and borderline personality disorders [18,26,27] increase the risk of suicidal behavior.

Third, efforts have been made to identify reliable biological predictors of suicidal behavior, although the overall result is modest [28]. Low cholesterol levels were not reliable biological predictors [29,30]. Compared with healthy controls, first time suicide attempters had significantly higher concentrations of cholecystokinin (CCK; a 33-amino-acid peptide co-responsible for gastric emptying, gallbladder contraction, pancreatic enzyme release, and suppression of appetite) [4]. Compared with healthy controls, both current and recent suicide survivors had significantly lower oxytocin concentrations [7]. Oxytocin is a neuropeptide involved in stress regulation and the development and maintenance of social bonds [31,32]. As a general observation, oxytocin is closely involved in prosocial behavior [33]. As such, suicidal behavior could be considered a failure in the perception of the quality of the social world. More recently, Mullins et al. [34] identified a genetic risk locus that appeared to contribute to explaining shared underlying biological suicide attempts and further known risk factors such as sleep disturbances, risk-taking behavior, lower socioeconomic status, and poorer general health. 

Fourth, in support of the concept that suicidal behavior be understood as the result of biased and disturbed social cognition, several lines of research identified social impairment and the disconnection from social life [35,36,37,38,39,40] as a risk factor for suicidal behavior. Nazarzadeh et al. [41] identified social factors such as loneliness and weak family ties as stronger predictors of suicidal behavior than covarying economic factors. O’Connor and Nock [42] identified feelings of defeat, social rejection, entrapment, humiliation, and subjectively perceived low social support as particular contributors to the risk of committing suicide.

The fifth cluster has attracted some scientific interest, though research is scarce compared to the above and previously described clusters: attempting and committing suicide implies behavior-inducing cognitions such as sensation-seeking, pain tolerance, and lack of fear of death (Deshpande et al. [43]), including a state of overarousal [3,44]. In a study from 2005, among 1311 adult participants reporting a lifetime suicide attempt, almost half of them tried to cope with their suicidal crisis by overdrinking alcohol and about 33% with other forms of reckless behavior [45]. Next, specifically, sensation seeking has been associated with risk-taking behavior [46]. Among 2189 16- to 19-year old adolescents, higher scores for sensation seeking as a proxy of risk-taking behavior was associated with higher scores for suicidal ideation and suicide attempts [46]. However, surprisingly, to our knowledge, the topic of risk-taking has not been further investigated among suicide survivors, and this holds particularly true for adult persons. Given this, the decision was to focus on this topic among adults. To this end, suicide attempters and healthy controls completed two self-rating questionnaires on suicidal behavior and risk-taking. 

Regarding the prevalence rates of suicides in Iran, data published in 2017 [47] from nationwide reports between 1991 and 2011 showed that the suicide rate was between 6.2 to 9.9 individuals out of 100,000 habitants, or between 3.6/100,000 for females and 7.0/100,000 for males [48]. The nationwide report further showed that the current suicide rate increased by 31 times between 1991 and 2011; such an increase was the highest in suicide-related deaths among the eastern Mediterranean region and among Islamic countries. To counter this, Iran’s national suicide prevention program (NSPP) was established in 2010 [49]. The interim report from 2021 [49] concluded that, contrary to previous data [47], the national incidence of suicide appeared to be low, with higher incidence rates among young adults, females, and in western regions. With this background in mind, it appeared reasonable to further investigate possible psychological factors to explain suicide attempts among Iranians. 

The two hypotheses and one research question were formulated as follows: First, based on observations among adolescents [46], we predicted that compared with healthy controls, suicide attempters reported higher scores for risk-taking. Second, we predicted that higher scores for risk-taking were associated with higher scores for suicidal behavior. With the research question, we asked in which specific dimensions of risk-taking behavior suicide attempters may differ from healthy controls. 

We hold that the present results may have the potential to add to the current literature in a specific though important fashion: is suicidal behavior related to risk-taking among adults?

## 2. Methods

### 2.1. Procedure 

Participants with SA and healthy controls (HC) were approached to participate in this study. They were fully informed about the aims of the study and the confidential and anonymous data handling. Thereafter, they provided the written informed consent. Next, participants completed three questionnaires covering sociodemographic information, suicidal behavior, and risk-taking behavior (see details below). Participants needed 20–30 min to complete the booklet. The ethical committee of the Kermanshah University of Medical Sciences (KUMS; Kermanshah, Iran) approved the study (ethics code: IR.KUMS.REC.1400.186), which was performed in accordance with the seventh and current version [50] of the Declaration of Helsinki. 

### 2.2. Participants 

A total of 616 participants took part in the study. Of those, 240 (39%; 22.1% females) reported a suicide attempt (SA) between one to 12 weeks ago (mean weeks: 7.9 weeks; SD = 2.55), and 376 (61%; 34.6% females) were healthy controls (HC) representing the healthy population (see more details below). 

#### 2.2.1. Participants with a Suicide Attempt

Individuals with objectively reported suicide attempts and admitted to the Farabi Psychiatric Hospital of the Kermanshah University of Medical Sciences (KUMS; Kermanshah, Iran) between 2019 and 2021 were consecutively included in the study. Inclusion criteria were as follows: 1. Aged 18 years and older; 2. Suicide attempt between one to 112 weeks ago, as ascertained by a trained and experienced psychiatrist or clinical psychologist and based on the medical records; 3. Willing and able to comply with the study conditions; 4. Provided written informed consent. Exclusion criteria were as follows: 1. Further acute suicidality and acute psychotic states, as ascertained by a trained and experienced psychiatrist or clinical psychologist and based on the structured clinical interview [51] for psychiatric disorders based on the DMS-5 [52]; 2. Withdrawal from the study. Of the 261 individuals approached, 240 (95.6%) agreed to participate. The main reason for nonparticipation was a further acute state of suicidality.

#### 2.2.2. Healthy Controls

To recruit healthy controls, the study was electronically posted on the intranet websites of the university hospital and university of Kermanshah (Kermanshah, Iran). Inclusion criteria were as follows: 1. Aged 18 years and older; 2. No past or current signs and symptoms of psychiatric issues, as ascertained by a trained and experienced psychiatrist or clinical psychologist and based on the structured clinical interview [51] for psychiatric disorders based on the DMS-5 [52]; 3. No suicide attempts in their life time; 4. Willing and able to comply with the study conditions; 5. Signed written informed consent. Exclusion criteria were as follows: 1. Withdrawal from the study. Of the 398 individuals approached and screened, 376 (94.5%) were included. Twenty-two either did not sign the written informed consent or did not complete the questionnaires. 

### 2.3. Measures

#### 2.3.1. Sociodemographic Information

Participants reported on their age (years), sex at birth (female; male), civil status (single; married; divorced), highest educational degree (compulsory school; diploma; higher diploma; bachelor or higher), and current occupation (unemployed; employed; housekeeping; student). 

#### 2.3.2. Translation Procedure

The Suicide Behaviors Questionnaire-Revised [53] and the Risk-Taking-18 questionnaire [54] were translated into Farsi. To this end, we followed the procedures as proposed by Brislin [55] and Beaton et al. [56]. First, two independent translators translated the English version into Farsi/Persian. Next, a third independent person compared the two translations, and, in case of differences, discussed the issues and performed the final draft. Next, two further independent translators performed the back-translation from Farsi into English and compared the back-translated English versions with the original version. Finally, the final version mirrored the general agreement of all five researchers involved in this procedure.

#### 2.3.3. Suicidal Behavior

Participants completed the Farsi version of the Suicide Behaviors Questionnaire-Revised (SBQ-R) [53]. The self-rating questionnaire consisted of four items, each tapping a different dimension of suicidality. The first item tapped into lifetime suicide ideation and/or suicide attempt; the second item assessed the frequency of suicidal ideation over the past twelve months; the third item assessed the threat of suicide attempts; and the fourth and last item evaluated the self-reported likelihood of suicidal behavior in the future. The four items had different patterns to answer on Likert scales ranging from 0 (=never) or 1 (=never) to 5 (=very often) or 6 (=very likely). The total sum score ranged from 3 to 18, with a higher score reflecting a more pronounced suicidal behavior. For the adult general population, the cut-off score of ≥7 reflects suicidal behavior, while for adults with diagnosed psychiatric disorders, the cut-off score of ≥8 reflects suicidal behavior (see [53]). We used the continuous and the categorical variable; for statistical comparisons, we used the more conservative cut-off value of ≥8 for both clinical and healthy samples. 

#### 2.3.4. Time Lapse between Suicide Attempt and Assessment

Participants with suicide attempts reported the time lapse in weeks between the suicide attempt and the current assessment. 

#### 2.3.5. Risk-Taking Behavior

To assess risk-taking behavior, participants completed the Farsi version of the Risk-Taking-18 (RT-18) questionnaire [54]. The self-rating questionnaire consisted of 18 items. Typical items were “I sometimes do ‘crazy’ things just for fun”; “I often follow my instincts, hunches, or intuition without thinking all the details”; or “I like “wild” uninhibited parties”. Answers were given on dichotomous scales—no = 0 and yes = 1—with higher sum scores reflecting a more pronounced risk-taking behavior. While de Haan et al. [54] proposed a two-factorial solution (risk-taking behavior, risk assessment), in the present study, we explored whether a further factor–solution could be identified. The decision was based on the assumption that a more fine-grained approach would also allow a more detailed assessment of the underlying psychological mechanisms associated with suicidal behavior and risk-taking. 

### 2.4. Analytic Plan

With a *t*-test and four X^2^-tests, we compared sociodemographic information (age, sex, civil status, highest educational level, employment status) between participants with SA and HC.

Next, to calculate the construct validity of the Farsi RT-18 (risk-taking) and SBQ-R (suicidal behavior), Cronbach’s alphas were calculated.

**Hypothesis** **1.***To compare the original dimensions of the risk-taking questionnaire (RT-18: Risk Assessment; Risk Taking Behavior; Risk Taking Total Score) between SA and HC, a multivariate ANOVA was performed with Risk Assessment, Risk Taking Behavior, and Risk Taking Total Score as dependent variables, Group (SA vs. HC) as factor, and age and gender as covariates*. 

Next, a factor analysis was performed to investigate the factor analytical properties of the RT-18. To this end, the 18 items were submitted to a factor with a Principal Component Analysis as Extraction Method and the Varimax with Kaiser Normalization as Rotation method. Following Brown [57], factor loadings > 0.40 were considered as meaningful. Excluded items: 8 and 11, as they showed high cross-factorial loadings.

The SBQ-R (suicidal behavior) suggests the cut-off value of ≥8 as a sign of high-risk of suicidal behavior. With a X^2^-test, we calculated the risk of reporting suicidal behavior (yes; no) between individuals with SA and HC. The odds ratio (OR) was also reported. 

Next, among SAs, the mean time lapse (weeks) between the suicide attempt and the current assessment was correlated with dimensions of age, suicidal behavior (SBQ-R), and risk-taking behavior (original and newly calculated factors of the RT-18). In the event of moderate to high correlation coefficients, the time lapse would have been introduced as covariate in the following calculations. 

Next, to compare the scores of suicidal behavior between participants with SA and HC, we performed a univariate ANOVA of the factor Group (SA, HC), with age and gender as possible confounders.

To compare the scores of risk-taking behavior between participants with SA and HC, we performed a multivariate ANOVA (new factors extracted from the RT-18) and the factor Group (SA, HC), with age and gender as possible confounders (Research question).

Effect sizes for *t*-tests were reported as Cohen’s d, and effect sizes for F-tests were reported as η_p_^2^.

**Hypothesis** **2.**
*As a last step, a series of Pearson’s correlations was performed between the new factors and the scores for suicidal behavior.*


All statistical analyses were performed with SPSS^®^ 28.0 (IBM Corporation, Armonk, NY, USA) for Apple Mac^®^. 

## 3. Results

### 3.1. General Information

Table 1 reports the statistical overview of sociodemographic information between SA and HC. Briefly, SA were prevalently female, married, with lower educational degrees and rather unemployed or housekeepers, or employed. HC were significantly younger, had more males than females, were generally single, reported higher educational degrees, and were prevalently students and/or unemployed.

### 3.2. Construct Validities; Cronbach’s Alphas

Cronbach’s alpha for the Farsi Suicide Behaviors Questionnaire-Revised (SBQ-R) was alpha = 0.83; Cronbach’s alpha for the Risk-Taking Questionnaire 18 (RT-18) was alpha = 0.76. Thus, based on the Cronbach’s alphas, the psychometric properties were satisfactory. 

### 3.3. Suicide Behavior (SBQ-R) between Participants with Suicide Attempts (SA) and Healthy Controls (HC)

Compared with HC (M = 5.97, SD = 3.42), individuals with SA (M = 12.68, SD = 5.15) reported statistically significantly higher scores (F(1, 612) = 291.74, *p* < 0.001, η_p_^2^ = 0.323 (large effect size), controlling for age and gender). 

### 3.4. Risk-Taking Behavior—Original Dimensions between Participants with Suicide Attempts (SA) and Healthy Controls (HC)

Table 2 provides the descriptive and inferential statistical overview of the RT-18 scores between individuals with SA and HC.

Compared with HC, individuals with AS reported higher scores for risk-taking behavior, risk assessment, and the total score (medium effect sizes), even when controlling for age and gender. 

### 3.5. Factor Analysis of the 18 Items of the Risk-Taking Questionnaire 18 (RT-18)

The 18 items of RT-18 were submitted to a factor analysis with a Principal Component Analysis as Extraction Method and the Varimax with Kaiser Normalization as Rotation method. Note that items were scored such that a higher score reflects a more pronounced pattern. 

Table 3 reports the factors and factor loadings of the items.

Four factors were extracted with Eigenvalues > 1. Overall, the factors explained 44.48% of the total variance. Factor 1 was labeled Thrill and Sensation Seeking, Eigenvalue: 3.133, explaining 19.58% of the total variance. Factor 2 was labeled Cautious Procedure, Eigenvalue: 1.724, explaining 10.773% of the total variance. Factor 3 was labeled Cautious Decision Making, Eigenvalue: 1.214, explaining 7.586% of the total variance. Factor 4 was labeled Impulsive Behavior, Eigenvalue: 1.047, explaining 6.541% of the total variance.

With regard to the factors separated by the SA and HC, the following significant mean differences were observed: SA showed the highest scores for thrill and sensation seeking (F(1, 614) = 63.78, *p* < 0.001, η_p_^2^ = 0.094 (medium effect size) and for impulsive behavior (F(1, 614) = 8.43, *p* < 0.01, η_p_^2^ = 0.064 (medium effect size), while HC showed higher scores for Cautious procedure (F(1, 614) = 37.26, *p* < 0.001, η_p_^2^ = 0.067 (medium effect size). For Cautious decision making, no statistically significant mean differences were observed (F(1, 614) = 2.13, *p* = 0.145, η_p_^2^ = 0.003 (trivial effect size). 

Overall, the four-factor solution revealed that SA had the highest scores for thrill and sensation seeking and impulsive behavior, while HC showed the highest scores for cautious procedure and decision making. 

### 3.6. Time Lapse between Suicide Attempt and Current Assessment

The mean time lapse in weeks between suicide attempts and the current assessment was 7.9 weeks (SD = 2.55). Correlation coefficients between the time lapse, age, dimensions of suicide behavior (SBQ-R), and risk-taking behavior (R-18: original factors—risk-taking behavior, risk assessment, risk-taking total score; newly calculated factors—Thrill and sensation seeking, cautious procedure, cautious decision making, impulsive behavior) were between −0.034 and 0.10; thus, correlation coefficients were spurious, and the time lapse was not further introduced as possible confounder. 

### 3.7. Risk of Suicidal Behavior between Participants with Suicide Attempt (SA) and Healthy Controls (HC)—Categorical Variables

Table 4 reports the descriptive statistics of high- (SBQ-R: ≥8) vs. low-risk suicidal behavior (SBQ-R: <8) between participants with SA and HC. Distributions of high- and low-risk suicidal behavior differed significantly between the three groups: Among SA, there were less participants with low risk and more with high risk than statistically expected. Among HC, there were more participants with low risk and less with high risk than statistically expected. The odds ratios (OR) to report high SBQ-R scores were 4.37-fold higher (CI: 3.47–5.50) in SA compared with HC. 

### 3.8. Associations between the New Factors of the Risk-Taking Questionnaire 18 (RT-18) and Suicide Behavior

A series of Pearson’s correlations was performed between the factor values and the scores for suicide behavior (SBQ-R). Higher suicide behavior was associated with higher thrill and sensation seeking (r = 0.36, *p* < 0.001), lower cautious procedure (r = −0.21, *p* < 0.001), lower cautious decision making (r = −0.26, *p* < 0.001), and higher impulsive behavior (r = 0.71, *p* < 0.001).

## 4. Discussion

The aims of the present study were to assess dimensions of suicidal behavior and risk-taking among individuals with a recent suicide attempt and healthy controls. Results showed that compared with healthy controls, suicide survivors (suicide attempters; SA) reported higher scores for risk-taking. More specifically, SA reported higher scores for thrill and sensation seeking and impulsive behavior. Thus, the present results expand upon the current literature in the following three ways: First, the association between risk-taking behavior and suicidal behavior has been observed among adolescents [46], and here, we showed this association among suicide survivors aged about 32 years. Second, among the range of risk-taking behavior, above all thrill and sensation seeking and impulsive behavior appeared to be risk factors. Third, dramatically, such risk-taking attitude appeared to persist even 1 to 12 weeks after a failed suicide. 

Two hypotheses and one research question were formulated, and each of these is considered now in turn.

With the first hypothesis, we assumed that compared with healthy controls (HC), suicide attempters (SA) would report higher scores for risk-taking, and data confirmed this. As such, we further confirmed what has been observed among 16- to 19-year old adolescents [46]. The add-on to the current literature is that such a pattern has been observed among suicide survivors aged about 32 years, and above all just one to 112 weeks after the suicide attempt. As such, though highly speculatively, one might suspect that among suicide attempters, the dimension of risk-taking might be either a rather stable personality trait, or the lack of alternative coping strategies, or both. 

With the second hypothesis, we predicted that higher scores for risk-taking were associated with higher scores for suicidal behavior, and data again confirmed this. 

With the research question, we asked in which dimensions of risk-taking behavior suicide attempters (SA) may differ from healthy controls (HC). To this end, we first performed a factor analysis of the Risk-Taking-18 questionnaire to identify a more fine-grained structure. It turned out that compared with HC, SA reported higher scores for thrill and sensation seeking and for impulsive behavior. Complementarily and not surprisingly, suicide attempters also reported lower scores for a cautious procedure to take decisions.

This abovementioned pattern of results demands particular attention. First, higher scores for sensation seeking and suicidal behavior have been observed among adolescents [46], even when controlling for depression and substance use. A meta-analysis revealed [58] a medium effect size between impulsivity and suicidal behavior. Further, a larger time gap between the moment of suicide attempt and assessment of impulsivity was associated with lower scores for impulsivity. In the present study, the assessment of risk-taking occurred 1 to 12 weeks after the suicide attempts. As such, it would have been conceivable that the assessment after a longer time frame night have yielded a more attenuated association. However, correlation coefficients of the time lapse between the suicide attempt and the current assessment (mean weeks: 7.9) and dimensions of suicidal behavior and risk-taking were spurious and trivial. As such, the present data suggest that suicidal behavior and risk-taking behavior do not appear to be related to a possible time-related recall bias or to the recency of the suicide attempts.

Next, two meta-analyses and systematic reviews revealed an association between suicidal behavior and deficits in inhibitory control and an excess in impulsive decision-making [59], and modest correlation coefficients between suicidal behavior and impulsivity (r = 0.19), aggression (r = 0.23), and impulsive aggression (r = 0.16) [60]. Further, higher scores for impulsivity mediated the relation between self-reported childhood maltreatment and suicidal behavior [61].

The question of whether and to what extent a suicide attempt is an impulsive and spontaneous action or, rather, the result of a premeditated and episodic planning is a complex question, and the answer to such a question appears to be complex, too. By nature, impulsivity implies a relative absence of premeditated planning. Opposite to this notion, results of a systematic review [62] showed two important results: First, effect sizes between trait impulsivity and suicidal behavior were small. Second, painful and fearful behaviors appeared to be the main driver for increasing the odds for suicide attempts. As such, episodic planning appeared to be more powerful to predict suicidal behavior, compared with trait impulsivity. Anestis, Soberay, Gutierrez, Hernández, and Joiner [62] concluded that painful, fearful, and provocative behaviors should be considered more thoroughly when dealing with predicting a suicide attempt. Overall, it appears that impulsivity and episodic planning might exclude each other. Here, we propose a theoretical framework to reconcile both dimensions: the theoretical model of motivation (Heckhausen’s motivational theory of motivation and behavior [63]) claims that a given behavior is the combination of longer lasting needs, motives, and goals (here, committing suicide) and a specific situation (place, timing, opportunity) to turn a goal into action [63]. As such, though highly speculatively, it is conceivable that a specific situational context may trigger the impulsive behavior to turn an episodic planning into action. Future studies might investigate if such a model based upon Heckhausen’s motivational theory of motivation and behavior [63] could be confirmed. 

The novelty of the results should be balanced against the following limitations. First, as extensively described in the Introduction section, suicidal behavior appears to be basically a disconnection from social life, and a social impairment as a result of biased and disturbed social cognitions. As such, it is highly conceivable that the cognitive–emotional elaboration of social interactions might have biased the present pattern of results. Of note, the claim of a dramatically impaired cognitive–emotional elaboration of social interactions was at least once neuroendocrinologically substantiated, such that compared with healthy controls, current and recent suicide attempters showed decreased oxytocin concentrations [7]. Second, the sample of suicide survivors was carefully selected regarding current psychiatric issues, including substance use disorder, borderline personality disorder, and mood disorders. However, everyday experience in clinical settings show that such “clean” samples do not reflect reality; again, to counter this, the association between risk-taking and suicidal behavior was not mediated by symptoms of depression and substance use, at least among a larger sample of adolescents [46], nor was suicidal behavior and risk-taking related to the time lapse between the suicide attempt and the current assessment. This pattern of results rather suggests that the risk-taking was a trait and not a state. Third, in our opinion, the motivational framework of behavior as a result between longer-lasting goals (here, committing suicide) and situational opportunities to turn a goal into action [63] deserves more attention, as it might help to overcome the apparently irreconcilable dichotomy between episodic planning and impulsivity. Last, while research on the biological basis of suicidal behavior appears highly unconclusive, Mullins et al. [34] reported to have identified specific genetic factors, which might confer to an increased risk of suicide attempts. 

## 5. Conclusions

Compared with healthy controls, recent suicide survivors reported higher scores for risk-taking in general, and for thrill and sensation seeking and impulsive behavior, specifically. Further, such risk-taking behavior was associated with persistent suicide behavior 1 to 12 weeks after a failed suicide. It appeared that suicide survivors might need particular counseling and support immediately after a suicide attempt. Further, suicide survivors might benefit from specific trainings to improve their social and emotional competencies.

## Figures and Tables

**Table 1 jcm-11-04177-t001:** Statistical overview of sociodemographic information between participants with suicide attempts (SA), and healthy controls (HC).

	Groups		Statistics
	Suicide attempters	Healthy controls	
N	240	376	
	M (SD)	M (SD)	
Age (years)	31.47 (7.71)	24.27 (5.70)	t(614) = 13.30, *p* < 0.001, d = 1.10
Time lapse (weeks)	7.90 (2.55)	-	-
	ns	ns	
Sex (male; female)	187/53	246/130	X^2^(N = 616; df = 1) = 10.95, *p* < 0.001; AS had more males than statistically expected; HC had more females than statistically expected
Civil status (single; married; divorced)	139/86/15	308/68/0	X^2^(N = 616; df = 2) = 53.59, *p* < 0.001; AS: less singles, more married, and more divorced than statistically expected; HC: more singles, less married, and less divorced than statistically expected
Highest educational degree (compulsory school; diploma; higher diploma; bachelor or higher)	98/90/31/21	144/130/90/104	X^2^(N = 616; df = 3) = 5.96, *p* = 0.113; descriptively: AS: more diploma and less higher diploma and bachelor than statistically expected; HC: less underdiploma and diploma, and more higher diploma and bachelor than statistically expected
Current occupation (unemployed; employed; housekeeping; student)	31/110/28/71	274/39/38/25	X^2^(N = 616; df = 3) = 232.29, *p* < 0.001; AS: much less students and more unemployed, housekeepers, and employed than statistically expected; HC: much more students, more housekeepers, and much less unemployed and employed than statistically expected

**Table 2 jcm-11-04177-t002:** Descriptive and inferential statistical indices of Risk-Taking 18 (RT-18) between participants with suicide attempts (SA) and healthy controls (HC).

	Groups		Statistics
	Suicide attempters	Healthy controls	
N	240	376	ANOVA
	M (SD)	M (SD)	
Risk-Taking			
Risk-Taking Behavior	5.29 (2.05)	3.93 (2.40)	F(1, 612) = 37.52; *p* < 0.001; η_p_^2^ = 0.071 (medium effect size)
Risk Assessment	4.67 (1.87)	3.61 (1.92)	F(1, 612) = 32.63; *p* < 0.001; η_p_^2^ = 0.072 (medium effect size)
Risk-Taking Total score	9.98 (3.12)	7.54 (3.54)	F(1, 612) = 75.88; *p* < 0.001; η_p_^2^ = 0.101 (medium effect size)

Covariates: gender (male; female) and age.

**Table 3 jcm-11-04177-t003:** Factor analysis of the Risk-Taking-18 (RT-18) questionnaire.

Factors	Item	Factor 1	Factor 2	Factor 3	Factor 4
Item Risk-Taking		Thrill/Sensation Seeking	Cautious Procedure	Cautious Decision Making	Impulsive Behavior
Do you enjoy taking risks?	4	0.570			
Would you enjoy parachute jumping?	5	0.536			
Do you welcome new and exciting experiences and sensations, even if they are a little frightening and unconventional?	6	0.630			
I often try new things just for fun or thrills, even if most people think it is a waste of time	7	0.511			
I enjoy getting into new situations where you cannot predict how things will turn out	14	0.647			
I sometimes like to do things that are a little frightening	15	0.625			
I sometimes do “crazy” things just for fun	16	0.633			
I prefer friends who are excitingly unpredictable	17	0.526			
Do you often get into a jam because you do things without thinking?	1		0.696		
Do you usually think carefully before doing anything?	2		0.666		
Do you mostly speak before thinking things out?	3		0.582		
I like to think about things for a long time before I make a decision	9			0.839	
I usually think about all the facts in detail before I make a decision	10			0.665	
I often follow my instincts, hunches, or intuition without thinking through all the details	12				0.651
I often do things on impulse	13				0.647
I like “wild” uninhibited parties	18				0.556

**Table 4 jcm-11-04177-t004:** Descriptive statistic of high- (SBQ-R: ≥ 8) vs. low-risk suicidal behavior (SBQ-R: <8) between participants with substance use disorder (SUD), attempted suicide (AS), and healthy controls (HC).

	Risk Level of Suicidal Behavior		Statistics
	Low Risk	High Risk	
	n	n	X^2^(N = 616, df = 1) = 212.29, *p* < 0.001
Suicide attempters (SA)	56	184	
Healthy controls (HC)	310	66	

## Data Availability

Data are provided to scientific researchers with expertise in the field; such scientific researchers should provide a detailed plan of hypotheses and a thorough explanation, as to why data are requested.

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
