# Peer review of "Risk-Taking Behavior among Suicide Attempters"

_jcm, 2022, doi:10.3390/jcm11144177_

Round 1

Reviewer 1 Report

The manuscript "Risk-taking behavior among suicide attempters" has an interesting objective, which is to determine if impulsive behavior accounts for the suicide risk.  However, there is too much information that is missing, and an important flaw in planning in this case-control study.  The manuscript shows some results of 240 individuals who had attempted suicide and 376 healthy controls. In the suicide attempt, there were 77% females and in the controls, the females were 34% of the group. The ages are not shown in spite of their importance to the epidemiology of suicide. And taking the information of the patients 1 to 3 months after the suicide attempt gives a very wide range of time after the event, which may compromise recall, mood changes, and the way the individuals report their behaviors. Because there are so many factors influencing the risk of suicide, such as gender and time of the year, the controls should have been paired with the suicide attempters for sound comparison.    I suggest the introduction brings information about the differences in suicide ideation, suicide attempt, and suicide and their possible interplay with impulsivity. A review of these interactions would be informative for the reader.  In methods, it would be important to separate the analysis related to the validation of the scales and the analysis regarding impulsivity and suicidal behaviors. It would be interesting to see tables with the sociodemographics of the two groups and the bivariate analysis of the factors followed by multivariate analysis. English style needs much improvement. 

Author Response

We thank Reviewer #1 very much for all their kind efforts and valuable comments, which helped us to improve the quality of the manuscript. Please find attached the revised file, along with the detailed point-by-point-response.

Again, thank you very much for the care devoted to the present manuscript.

Reviewer 2 Report

This is an interesting paper. There is a lot more to learn on associations and predictors of suicide behavior and risk taking is potentially one of these factor but research in this area is quite scarce It is an important topic, because the latest research in Genetics show that there is an independent gene cluster (independent from depression) associated with risk seeking and suicidality: “Dissecting the Shared Genetic Architecture of Suicide Attempt, Psychiatric Disorders, and Known Risk Factors” Biological Psychiatry 2021. Given this discovery is it very interesting to gain some more knowledge on this topic. However, I feel that the paper can be improved. I have the following comments.

Introduction

1)      I would integrate ( a paragraph) this gene finding (see above) somewhere in the introduction (P2, R53) (or discussion). It creates a better context and it points out the value of the findings in this study more clearly.

2)      P1, R43. As this is about the Iranian population please add some information also  on the severity of suicide in Iran (xxx cause of death in Iran)

3)      I do not think that social cognition is biased for all suicide attempters p2, R70). Biased means systematic chronic deviations. I would prefer the term disturbed or both biased or disturbed. Disturbed gives an indication that treatment may help to normalize this.

4)       I agree with the authors that this research in the 5th cluster is scarce, but it is not non-existent among adults. P2, 83. Leo D, et al. J Affect Disord. 2005. PMID: 15935241.. Fore example. My suggestion would be to mitigate the text.

Method

Participants (P3, r107-123)

We need more information on the sampling frame and sampling procedures. Now we cannot understand how these results are representative of the populations mentioned. Were the suicide attempters selected in mental health care, general practitioners, community.? The same goes for the healthy controls: were they matched?  Probably not because there are a lot of differences in socio demographics in the results.

How was decided that these numbers were enough (power analysis) to detect differences?

There is no statement on nonresponse (or figures)

Measurements

Nicely written!

P4 R153 (suicidal behavior)

Could you indicate the value of this threshold with a reference or previous research in your population? If that is not existing than please explain why -you as a research group- used these thresholds (of 7 and 8).

Analytic plan

It would help to make a connection between the hypothesis in the introduction and the analytic plan. I had to think again about why an explanatory factor analysis was needed. (p4, r170)

Results

There are quite a lot of statistically significant differences on Socio-Demographics. Despite these differences, the SA persons show more Risk Seeking.

This is interesting because Risk seeking behavior is more often found in males and surviving a suicide attempt is more often found in females.

I think the article will gain value when the analyses  in 3.5 (P7, r248) are done in a slightly different way. One should incorporate this sociodemographics in the model that explains the differences between SA and HC. Probably a stepwise regression is better and showing the differences between a sparse model (only the difference between HC and SA) and a model in which you control this difference while controlling for sociodemographic differences would be very appealing for the reader.

Discussion and Conclusion

No comments! Good overview and bridge to treatment opportunities

Author Response

We thank Reviewer #2 very much for all their kind efforts and valuable comments, which helped us to improve the quality of the manuscript. Please find attached the revised file, along with the detailed point-by-point-response.

Again, thank you very much for the care devoted to the present manuscript.

Round 2

Reviewer 1 Report

Thank you for the changes made in your manuscript it is much improved; the language is appropriate and the text brings a clear message. The differences in the translation of the instrument and its results have been clarified. 

There are some sparse typos in  " line 62 cholezystokinin" and  Line 246 - correct "singles". I believe that in "Line 144 "of them"  is not necessary".  

Author Response

Dear Reviewer,

Thank you again for all your kind efforts.

We have addressed all concerns raised by the Reviewers. Please see the detailed point-by-point-response attached as a separate file. 

Again, thank you very much for the care devoted to our manuscript. 

Reviewer 2 Report

I think the authors did a good job in explaining the study and statistics better. It is also better framed in the context of recent literature. It resulted in an interesting paper and adds to the scarce literature on risk seeking behavior and suicideology

Author Response

Dear Reviewer,

Thank you again very much for all your kind efforts.

We have addressed all concerns raised by the Reviewers. Please see the detailed point-by-point-response attached as a separate file. 

Again, thank you very much for the care devoted to our manuscript. 
